

# Benzo(a)pyrene in Urban Environment of Eastern Moscow: Pollution Levels and Critical Loads

Nikolay S. Kasimov, Elena M. Nikiforova, Natalia E. Kosheleva, Dmitry V. Vlasov

Department of Landscape Geochemistry and Soil Geography, Faculty of Geography, Lomonosov Moscow State University,
Moscow, 119991, Russian Federation

*Correspondence to*: Dmitry V. Vlasov (vlasov.msu@gmail.com)

**Abstract.** Polycyclic aromatic hydrocarbons (PAHs) and particularly benzo(a)pyrene (BaP) are toxic compounds emitted from various anthropogenic sources. Understanding the BaP concentrations, dynamics and decomposition in soil is required to assess the critical loads of BaP in urban environments. The first attempt to evaluate all major input and output components
of benzo(a)pyrene (BaP) balance and to calculate the permissible load on the urban environment in different land-use zones of the Eastern District of Moscow was done. BaP contamination of the snow cover in the Eastern District of Moscow was related to daily BaP fallouts from the atmosphere. In 2010 the mean content of the pollutant in the snow dust was 1942 ng·g$^{-1}$ whereas the average intensity of its fallout was 7.13 ng·m$^{-2}$ per day. Across the territory BaP winter fallout intensities varied from 0.3 to 1100 ng·m$^{-2}$ per day. The average BaP content in the surface (0–10 cm) soil horizons was 409 ng·g$^{-1}$, which is 83
times higher than the local background value and 20 times higher than the maximum permissible concentration (MPC) accepted in Russia. The variations in soil and snow BaP concentrations among different land-use zones were examined. A significant contribution of BaP from the atmosphere to urban soils was identified. Based on the measurements of BaP atmospheric fallout and BaP reserves in the soils, the critical loads of BaP for the land-use zones of the Eastern District were calculated for different values of degradation intensity and different exposure time. It was established that, at annual
degradation intensity of 1–10 %, the ecologically safe BaP levels in soils of all land-use zones, excluding the agricultural one, will be reached only after many decades and centuries.

## 1 Introduction

The early 21$^{st}$ century is characterized by a significant exacerbation of environmental problems, especially in large industrial and urban centers. According to demographic projections by 2050 about 66 % of the 9 billion people on Earth will be living
in cities (UN DESA, 2015), which will lead to an increase in anthropogenic impact on the environment and its further contamination.

Toxic substances, like polycyclic aromatic hydrocarbons (PAHs) and their most dangerous member with carcinogenic and mutagenic properties – benzo(a)pyrene (BaP) – are of significant importance in environmental studies (Haglund et al., 1987; Gennadiev and Pikovskii, 1996; Garban et al., 2002; Chung et al., 2007; Jacob, 2008; Pergal et al., 2015). This gives a



special relevance to a research focused on long-term dynamics of BaP contents in urban environment related to the intensity of BaP emissions from anthropogenic sources on the one hand and its accumulation and destruction in soils on the other hand.

BaP is released from industrial, heating, vehicle and domestic wastes. It is also a byproduct of organic waste and fuel
combustion (Larsen and Baker, 2003; Pergal et al., 2015). BaP is added to urban environment from polluted air with dust, precipitation and aerosols and accumulates in surface layers of soils (Wania and MacKay, 1996; Trapido, 1999; Fernández et al., 2000; Nam et al., 2009). In the United Kingdom, over 90 % of all PAHs reserves in the environment are limited to the surface layer of soil (Wild and Jones, 1995). PAHs concentration in the air is generally significantly higher in winter than in summer because of greater fuel combustion (Ollivon et al., 2002; Gaga et al., 2009; Birgül et al., 2011).

In many major cities of the world BaP concentrations in the environment are ten to hundred times higher than the regional values (Trapido, 1999; Ma et al., 2005). Long-term (1990–2006) monitoring of BaP concentrations in the Eastern Administrative District (EAD), one of the most polluted parts of Moscow, demonstrates a steady increase in pollution and related deterioration of soil functioning (Kosheleva and Nikiforova, 2011).

The rate of BaP accumulation in soil depends on the balance between its fallout from the atmosphere and the intensity of
removal and decomposition. BaP addition to urban soil is almost entirely derived from anthropogenic sources, whereas BaP removal through volatilization, degradation and leaching depends on landscape, geochemical and bioclimatic factors (Morillo, 2008; Kosheleva and Nikiforova, 2011). BaP input is measured by the rates of its deposition with road and industrial dust (Yu et al., 2014). In cities with cold climate, the BaP deposition from the atmosphere can be determined by examining BaP reserves in the snow cover (Haglund et al., 1987; Kasimov, 1995; Sharma and McBean, 2001). However,
particulate matter or water soluble fraction of snow are rarely analyzed for BaP contents, so the contribution of these fractions to the formation of BaP anomalies in urban environment has been yet poorly studied.

Among the individual PAHs, BaP is the least soluble in natural fresh waters and has limited ability to be leached from the soils. Therefore, the losses of BaP from the soils are mostly determined by its destruction which is enhanced by ultraviolet radiation (Gennadiev and Pikovskii, 1996). The rates of PAHs and BaP molecule transformation are also heavily dependent
on soil properties, such as pH, soil organic matter (SOM) content, particle aggregation, texture, heat and water fluxes and regimes, redox conditions, etc. PAHs and BaP biodegradation is affected by the molecular structure of specific compounds (Wild and Jones, 1995; Doelman, 1995; Kleeman et al., 2000; Johnsen and Karlson, 2007; Jiang et al., 2009; Birke et al., 2011; Kosheleva and Nikiforova, 2011; Kasimov et al., 2014).

Determination of BaP input and losses makes it possible to evaluate the resilience of urban soils to organic pollutants and to
assess the possibility of their self-purification. The concept of critical loads on urban environment provides a methodological framework for this kind of research. The critical load of a pollutant is defined as the maximal input to an ecosystem that does not lead to irreversible changes in its biochemical structure, biodiversity and the productivity over a long period of time (Vries et al., 1997; Baskhin et al., 2004). So, the aim of the present study is to analyze the fate of BaP in urban environment of the EAD of Moscow using the concept of critical loads. The particular tasks are as follows:





- to identify the intensity of atmospheric BaP addition to the snow cover in winter period on the basis of dust deposition rates and the BaP contents in the solid fraction of snow;

- to determine BaP concentrations and to reveal the specific features of BaP spatial distribution in soil cover of EAD, including the localization and size of anthropogenic anomalies;

- to assess the ecological risk of BaP pollution on the basis of sanitary and hygienic standards for BaP concentration in soils, accepted in Russia by legislation;

- to calculate BaP critical loads on urban soils with respect to BaP degradation rates and exposure period.

## 2 Materials and Methods

### 2.1 Study area

The work was conducted on the territory of Moscow, one of the largest cities in the world with a population of about 12 million. The city of Moscow is divided into twelve administrative districts. The study area is located in the Eastern part of Moscow which is considered to be one of the most polluted in the city because of its large industrial sector and concentration of large processing enterprises. The study area belongs to the southern taiga landscapes of the Meschera Lowland which is a flat outwash plain with mean altitudes of about 150 m (Fig. 1). The land surface is dominated by urban structures. Urbic

Technosols dominate in the soil cover; their morphological, physical and chemical properties differ greatly from those of the background Retisols developing under coniferous-deciduous forests on loamy deposits (Kosheleva and Nikiforova, 2011). Anthropogenically modified and artificial soils are also represented by Eutric Retisols, Transportic Technosols, Ekranic Technosols (sealed by asphalt) and by few other soil types developed on man-made sediments and the cultural layer (Prokof'eva et al., 2014).

Traffic releases more than 90 % of pollutants in aerosol and gase phase, including PAHs, into the air and thus defines the ecological situation in the District (Kasimov et al., 2014). Car exhaust emissions from large highways cause a significant adverse effect on urban soils (Fig. 1). A few large industrial zones with chemical and petrochemical plants and two thermal power plants are also sources of BaP pollution.

### 2.2 Sampling

The snow cover in the EAD was surveyed in early March, when the snow depth was maximal, while the soil sampling was carried out in June 2010 using the methods recommended in (Revich et al., 1982; Kasimov, 1995). Land-use zoning was accomplished and traffic, industrial, residential, recreational and agricultural zones were distinguished (Nikiforova et al., 2014). Overall, 50 snow and 50 topsoil (0–10 cm) samples were collected at the same locations at a regular grid with 800m intervals. Composite snow samples were formed by mixing 10 replicate samples taken with a plastic tube 5 cm in diameter.

Soil samples were compiled from 3–4 individual ones. Due to a prevailing western atmospheric transfer, background (reference) snow samples were collected 50 km west of Moscow, near the city of Zvenigorod. The background area for



urban soils was located in the Meschera Lowland, 40–50 km east of Moscow, since the natural soils in that area (Retisols) were formed on the same parent materials (sandy clays and light clay loams) as the urban soils.

## 2.3 Laboratory technique and analysis

The solid fraction (particulate matter) was separated from snow samples using a membrane filter with a pore diameter of

0.45 μm. BaP contents in the snow solid fraction and in the bulk soil samples were measured via low-temperature spectrofluorimetry under Shpolsky effect using "Fluorat-Panorama" complex (Lumex, Russia), which was supplemented by "LM-3" monochromator and "CRYO-1" cryogenic attachment, with precision of ±25 % (Alekseeva and Teplitskaya, 1981). Geochemical data treatment included the calculation of the enrichment factor $EF$ which compares the abundances of BaP in the urban soils to the reference value. For the assessing the risk of contamination the ecological risk index $PI$ was estimated

on the basis of maximum permissible concentration (MPC) which is a general BaP sanitary level equal to 20 ng·g$^{-1}$ (GN.2.1.7.2041-06). The intensity of BaP atmospheric fallout, ng·m$^{-2}$ per day, is calculated as follows: $D = P_n \times C$, where $P_n$ is the daily dust deposition on the urban territory (g·m$^{-2}$) and $C$ is the BaP concentration in the solid fraction of snow (ng·g$^{-1}$); $P_n = m/(n \cdot l \cdot s)$, where $m$ – mass of solid matter in a snow sample, ng; $n$ – number of tubes with snow collected in sampling points; $l$ – number of days with snow cover; $s$ – cross-section area of tube, m$^2$. The excess of BaP atmospheric fallout over

background level is defined as $LF = D/D_b$, where $D$ and $D_b$ refer to the BaP fallout intensities on the urban and background areas, respectively.

## 3 Results and Discussion

### 3.1 BaP concentrations in snow and its deposition from the atmosphere in background and urban environments

BaP is commonly added to soil cover through atmospheric precipitation falling either in liquid or solid phases. Since the

velocity of snowfall is slower and snowflakes have larger surfaces compared to raindrops, they become rich in BaP faster (Vasilenko et al., 1985).

In the *background area* near Zvenigorod the snow is weakly acidic (pH 5.8). The average BaP content in the dust fraction of snow equals 169 ng·g$^{-1}$ (Table 1), which denotes low atmospheric addition of BaP to the background soils. BaP distribution in the snow cover is characterized by high spatial variability with a variation coefficient ($C_v$) of 106 %. BaP fallout intensity

is estimated to be approximately 2 ng·m$^{-2}$ per day, which is close to the values 0.1–2.2 ng·m$^{-2}$ per day, reported earlier for the unpolluted area in the Serpukhovsky District of Moscow Region by Chernyansky et al. (2007), measured by passive sampling.

The estimations performed by Gabov et al. (2008) for the undisturbed territories of the East European Plain in the taiga zone show that the annual average rate of BaP deposition with dust and snowfall to the soil surface is 30–40 ng·m$^{-2}$. These

elevated fallout values were explained by a large atmospheric concentration of BaP due to high anthropogenic emissions on





a regional level.

In this study, in *urban areas* snow water had a higher pH in relation to the background level. This is especially apparent in industrial zones (pH 7.6) and in residential areas with high-rise buildings outside of the Moscow Ring Road (pH 6.4). The solid fraction of snow in the EAD was enriched with BaP: its average content was 1942 ng·g$^{-1}$, which is 11.5 times higher than the background level. BaP contents in the snow differed dramatically across the land-use zones (Table 1). They can be arranged according to their corresponding enrichment factor (*EF*) values in the following order: industrial (*EF*=34) > traffic (21) > mid-rise residential blocks (7) > low-rise quarters (4) > high-rise blocks (1.3) > agricultural (1.2) > recreational (0.9). The BaP content in the solid fraction of snow in particular land-use zones varied within a wide range: $C_v$ exceeded 100 % in 5 out of 7 zones (Table 1).

BaP delivery from the atmosphere is defined not only by its content in solid particles of snow, but also by their fallout intensity. In the EAD, the BaP fallout with the snow dust during the winter period was quite significant and it averaged at 75.6 ng·m$^{-2}$ per day, ranging from 189 in industrial zone to 2.4 in the recreational zone (Table 1).

BaP fallouts in the two most polluted zones were much higher than in the background territories (*LF*=93–80) however the difference between the recreational zone and the reference sites was insignificant (*LF*=1.2). These data on the intensity of BaP fallouts with snow dust were consistent with the results reported for the city of Bratsk, Siberia. In the industrial zone of Bratsk, where a large aluminum plant and a pulp and paper factory are located, the BaP load rises up to 1000 ng·m$^{-2}$ per day, while in the residential area of the Central District it was 100–200 ng·m$^{-2}$ per day (Kasimov, 1995).

Mean anthropogenic BaP addition to soil cover in Moscow's EAD (75.6 ng·m$^{-2}$ per day) is comparable to fallout levels in other urban territories of the world with industrial and transport emissions of BaP. For example, in Bursa, Turkey, the daily BaP deposition equaled 100–1000 ng·m$^{-2}$ (Esen et al., 2008, Birgül et al., 2011); in Ankara it was 65 ng·m$^{-2}$ in spring and winter and 10 ng·m$^{-2}$ in summer and autumn (Gaga et al., 2009). In Jersey City, NJ, USA, the daily BaP addition with precipitation made up 25 ng·m$^{-2}$ and dry deposition accounted for 83 ng·m$^{-2}$ (Gigliotti et al., 2005). In Nahant, MA, USA, BaP fallout increased up to 71 ng·m$^{-2}$·day$^{-1}$ (Golomb et al., 1997), while in Kuopio, Finland, the fallout ranged from 40 to 250 ng·m$^{-2}$·day$^{-1}$ (Hautala et al., 1995).

## 3.2 BaP in background and in urban soils

The surface horizons of the background soils, which are represented by Retisols, were acidic (pH 5.5) and low BaP concentrations, which were 34 times less than the mean BaP value in the solid fraction of snow (Table 1). Still, like dust in the snow cover, the natural soils show typically high variability of BaP contents ($C_v$=92.7 %). For example, undisturbed Norwegian soils have the same level (5 ng·g$^{-1}$) of BaP content but the world's background is characterized by higher BaP values – 22 ng·g$^{-1}$ (Nam et al., 2009).

*Urban soils* are weakly alkaline, ranging in the reaction from basic (pH 8.1 in industrial zone) to weakly acidic (pH 6.5 in recreational zone). Based on our data, the average BaP concentration was 409 ng·g$^{-1}$, which is 43 times higher than the background level. The BaP content varied greatly across land-use zones (Table 1), reaching a maximum in the industrial





zone and along highways, where the surface horizons represent well-defined anthropogenic geochemical barrier (Kosheleva et al., 2015). According to the BaP enrichment of the surface layers the land-use zones can be arranged in the following order: industrial ($EF$=241) > large highways (115) > mid-rise residential (72) > recreational (66) > low-rise quarters (19) > high-rise blocks (11) > agricultural (0.9). BaP concentrations in the urban soils vary greatly: its variation coefficients $C_v$

exceed 130 % in 5 out of 7 zones.

In the majority of land-use zones the solid fraction of snow had higher (4–7 times) BaP concentrations than the surface soil layers. This situation is typical for the case, when relatively high BaP concentration is observed for the airborne particles (Amagai et al., 1999). In the agricultural zone the relative enrichment of snow dust with BaP increased up to 42 times with respect to soil material. The difference between BaP content in the snow and in the soils can be attributed to two factors.

First, the nearby sources, such as Kozhukhovo settlement, where homes are individually heated with furnaces, and the Rudnevo industrial zone with a waste incineration plant probably released a considerable amount of BaP. Second, the cultivated soils of this zone had higher BaP decomposition rates compared to the soils in other zones due to favorable moisture and air regimes (Nikiforova and Alekseeva, 2005). The ratio between the average BaP concentrations in snow and in the surface soil horizons indicated a significant BaP addition from the atmosphere and its progressive accumulation in the

soils especially in the industrial and traffic zones.

In order to evaluate the ecological risk associated with BaP soil pollution, the average concentrations of BaP in the District's land-use zones were compared with the sanitary standards (MPC) accepted in Russia. The highest risk index *(PI)* was revealed in the industrial zone (*PI*=78), followed by traffic zone (28) and residential areas with mid-rise buildings (14). The lowest level of the risk was observed in the residential high-rise (1.7) and the agricultural (0.2) zones outside the Moscow

Ring Road. The average BaP content in the urban soils exceeds the norm by 20 times, indicating that the soils are in an extremely dangerous ecological condition with regards to the BaP pollution. Territories where the BaP content is 30 times or more higher than the MPC make up 20 % of the District's area, whereas 35 % of the area is occupied by the soils with the BaP concentration being 10–30 times higher than the MPC. The rest of the EAD territory is occupied by clean soils and soils with the MPC exceedance no more than 10 times.

The measured BaP contents in soils of Moscow's EAD are much higher than the BaP levels recorded in the majority of the cities in the world (Table 2), however the comparison of our data with the results of BaP surveys in Moscow, as well as in a number of Russian and Belarusian cities indicates similarity in the BaP levels. As a rule, the average BaP concentrations in the soil and in the atmospheric fallout make up no more than 10–15 % of the total PAHs content (Garban et al., 2002) and maximum concentrations are usually observed in the soils in the industrial and traffic zones.

**3.3 Critical loads of BaP**

In order to determine the critical loads of BaP on urban soils of the various land-use zones of EAD, the topsoil reserves of BaP were calculated. In the calculation we considered the uppermost soil layer with a thickness of 10 cm and an area of 1 m$^2$ and the BaP concentrations ($C,$ mg·t$^{-1}$) measured in 2010. Then the topsoil reserves of BaP equal $B = S \cdot h \cdot \rho \cdot C$, where $B$ is the





BaP reserve in the surface layer of soil in mg, $S$ is the model area (1 m$^2$), $h$ is the thickness of the soil horizon (0.1 m) and $\rho$ is its density in t·m$^{-3}$. The density of urban soils varies from 1.12 to 1.93 t·m$^{-3}$ (Charzyński et al., 2013). Density is one of the diagnostic characteristics of the technogenic horizon of urban soils: it cannot be less that 1.2 t·m$^{-3}$ (Prokof'eva et al., 2014). Therefore, we took the soil density to be 1.2 t·m$^{-3}$ in recreational and agricultural zones and 1.4 t·m$^{-3}$ in all the other zones

which are subjected to significant anthropogenic impact. The topsoil reserves ($B$, mg) in different land-use zones decrease in the following sequence: industrial (219) > traffic (79.2) > mid-rise residential (39.6) > recreational (19.7) > low-rise residential (11.8) > high-rise residential (4.84) > agricultural (0.564).

The MPC of 20 ng·g$^{-1}$ was used to define the critical level of BaP in urban soils below which significant harmful effects on the soil biota do not occur. Critical BaP reserves in the soil surface horizons ($B_{MPC}$, mg) correspond the MPC of BaP in

topsoils. They are 2.4 mg in agricultural and recreational zones and 2.8 mg in others.

Because the BaP concentrations exceed the MPC in the majority of the EAD land-use zones, we considered a load as critical if after a specific time interval (5, 10, 25 years, etc.), and given the rate of pollutant's degradation, the BaP content would decrease to the permissible level.

The long-term dynamics of BaP reserves in the surface soil horizon ($B$, mg) can be described by an Eq. (1):

$$\frac{dB}{dt} = -\beta \cdot B + v, \tag{1}$$

where $\beta$ is the rate of BaP degradation (year$^{-1}$) and $v$ is the intensity of its atmospheric fallout (mg) per 1 m$^2$ per year. This equation assumes that the process of BaP degradation in soil follows a first-order kinetic model and that BaP input is constant. BaP vertical and lateral off-site translocation was ignored, because practically all BaP is absorbed on soil particles (Gennadiev and Pikovskii, 1996; Kosheleva and Nikiforova, 2011).

Equation (1) can be solved by dividing the variables:

$$-\frac{1}{\beta} \ln(v - \beta \cdot B) = t + \alpha_0,$$

where $\alpha_0$ is a random constant. Transforming this equation gives Eq. (2):

$$v - \beta \cdot B = e^{-\beta \cdot \alpha_0} e^{-\beta t} = \alpha_1 e^{-\beta t}, \tag{2}$$

where $\alpha_1 = e^{-\beta \cdot \alpha_0}$. Initially, when $t=0$, $v - \beta \cdot B = e^{-\beta \cdot \alpha_0} = \alpha_1$, $B=B_0$, and Eq. (2) takes the form $v - \beta \cdot B_0 = \alpha_1$.

Taking into account the initial condition we get the solution of the Eq. (1):

$$B = \frac{v}{\beta} - (\frac{v}{\beta} - B_0)e^{-\beta t}.$$

With an annual degradation intensity of $\beta$ and a BaP fallout rate equal to the critical load $v = D_{MPC}$, the critical reserve of

BaP ($B_{MPC}$) in the surface layer of soil equals to: $B_{MPC} = \frac{D_{MPC}}{\beta} - (\frac{D_{MPC}}{\beta} - B_0)e^{-\beta t}$ or



$$B_{MPC} = \frac{D_{MPC}}{\beta}(1 - e^{-\beta t}) + B_0 e^{-\beta t}.$$

Then the BaP critical load is determined by the Eq. (3):

$$D_{MPC} = \beta \, (B_{MPC} - B_0 \cdot e^{-\beta t}) \, / \, (1 - e^{-\beta t}) \tag{3}.$$

The intensity of the relative BaP degradation is given in fractions of 1. For example, the degradation intensity of 1 % means $\beta = 0.01$, 5 % means $\beta = 0.05$, etc; $t$ is the model time of exposure in years. BaP can be preserved in soils for many years and decades, and even for longer time periods: hundreds and thousands of years (Shilina et al., 1980; Wild et al., 1991; Gennadiev et al., 2004).

According $\beta$ estimates reported in (Wild and Jones, 1995), the annual degradation intensity in soils is 10 % for PAHs and 6 % for BaP. In order to take into account the variability of the BaP contents in the soils related to different soil conditions, its critical loads were calculated using several values of degradation rates, ranging from 1 % to 10 % (Fig. 2).

The analysis of Fig. 2 demonstrates that the critical loads ($D_{MPC}$) vary greatly depending on the BaP degradation rates and the exposure time in all studied zones except for the agricultural one. If the BaP decomposition is assumed to be slow (1–3 % yearly), the soil pollution in most of the land-use zones will decrease to the permissible level only under insignificant BaP addition and during long exposure time (over 100 years). If the exposure time is shorter (even in case of nil atmospheric addition of the pollutant), the BaP reserves in the soil can't drop below the critical levels, which are 2.4-2.8 mg·m$^{-2}$, due to its low degradation rate. In the agricultural zone, the calculated critical loads of BaP are turned to be tens and hundreds times higher than the measured pollutant fallout rates because of initially low BaP levels in the soil.

As the BaP degradation rate increases, so does the value of the critical loads ($D_{PMC}$). However, even at a very high degradation intensity (10 % per year), the non-zero critical loads appear in the high-rise residential areas when the exposure period is longer than 10 years, in the low-rise residential areas – with the exposure period over 25 years and in the traffic, industrial and in the mid-rise residential zones – with the exposure time over 50 years, which can be explained by the high initial levels of BaP in these zones.

The critical BaP load values calculated using this model demonstrate that at a certain degradation intensity the BaP concentrations in the EAD soils can decrease to the level of MPC, although this will take a relatively long time. Based on the above formulas, the time necessary for BaP content in the soil to reach the MPC can be calculated as Eq. (4):

$$T = \ln[(B_0 - D/\beta) \, / \, (B_{MPC} - D/\beta)] \, / \, \beta \tag{4},$$

where $D$ is the BaP load measured in 2010, mg·m$^{-2}$ per day. The results of these calculations are shown in Table 3.

At existing levels of the BaP deposition with the insignificant degradation intensity (1–2 %), the BaP concentrations will reach the MPC in high-rise residential areas in 28–59 years. The BaP concentrations will lower to the MPC in the recreational zone in 106–214 years based on our calculations. It is worth to note that the MPC will never be reached in the other zones, because the input of BaP into the soils there exceeds its losses through the decomposition.



As the degradation intensity grows, the time period necessary for a decrease of BaP contents to the permissible level becomes shorter. At BaP degradation rates of 3–5% in traffic, industrial and low-rise residential zones soil BaP content may gradually decrease to the MPC after a few decades or centuries. The drop of the BaP content in the soils of the mid-rise residential quarters can only occur if the rate of degradation is 6 % or more. In the agricultural zone the BaP concentrations

can't exceed the permissible level even under low intensity of degradation because of insignificant addition of BaP from the atmosphere.

Thus, at the existing emission of BaP to the atmosphere and it subsequent accumulation to the soil, its content in the soils in all of the land-use areas, except for agricultural one, may come to the permissible level after decades and hundreds of years. In this situation, to achieve and to maintain environmentally acceptable BaP levels in urban soils one needs to remove the

contaminated soil surface layer and replace it with unpolluted soil material, and also to reduce the BaP emissions.

**4 Conclusions**

The intensity of the BaP fallout from the atmosphere to the urban soils with solid fraction of snow on the territory of EAD in Moscow ranged from 0.3 to 1150 ng·m$^{-2}$ per day. The average BaP content in the snow dust was 1942 ng·g$^{-1}$, which is 11.5 times higher than its level in the reference (background) sites. The BaP contents in the snow cover varied dramatically across

the land-use zones: from 5732 ng·g$^{-1}$ in the industrial zone and 3605 ng·g$^{-1}$ in the traffic zone to 154 ng·g$^{-1}$ in the recreational zone.

The average BaP content in the soil surface layers of Moscow's EAD in 2010 was 409 ng·g$^{-1}$, which is 83 times higher than the levels measured in the unpolluted background soils. The District's land-use zones can be arranged by the BaP soil enrichment in the following order: industrial > traffic > mid-rise residential > recreational > low-rise residential > high-rise

residential > agricultural. The BaP pollution on average exceeded 20 times the norm (MPC). The highest ecological risk posed by the BaP soil pollution was observed in the industrial zone and in the areas situated along large highways (*PIs* of 78 and 28 respectively) and the lowest risk was observed in the high-rise residential areas (*PI* =1.7) and in the agricultural zone (*PI* = 0.2).

The calculations of the BaP critical loads on the soils in the EAD showed that, at degradation intensities ranging from 1 to 10

% per year, the BaP concentrations may decrease to the permissible level only after a relatively long period of time – many decades and centuries. Because the urban soils have limited self-purification abilities, a combination of remediation measures must be taken to reduce the ecological risk posed by the BaP soil pollution.

The first experience of the application of the critical loads approach for BaP demonstrated its feasibility for contamination control of urban environment. It also showed the need for further research on BaP entry and loss rates from the urban soils,

especially in relation to the conditions and rates of the BaP decomposition. The reliable assessment of the pollutant's input and output is the basis for a sound prediction of the long-term dynamics of the contamination with the highly toxic organic compounds, such as BaP, in the urban environment.





**Author contribution**

All authors designed the experiments. E. Nikiforova, N. Kosheleva and D. Vlasov carried out fieldwork. D. Vlasov and
N. Koshleleva calculated critical loads of BaP. All authors calculated pollution levels of BaP in urban soils and snow cover
and all authors prepared the manuscript.

**Competing interests**

The authors declare that they have no conflict of interest.

**Acknowledgements**

This study was made with financial support from the Russian Science Foundation (project No. 14-27-00083). The authors
would like to thank the staff of the Department of Landscape Geochemistry and Soil Geography at the Faculty of Geography
of Lomonosov Moscow State University: N.I. Khlynina for her analytical work, as well as M.Y. Lychagin, I.N. Semenkov,
T.S. Koshovsky and G.L. Shinkareva for their participation in the field work. We are grateful to Dr. E.N. Aseyeva who
helped us to prepare the manuscript for publication. This work contributes to the Pan-Eurasian Experiment (PEEX) Program
research agenda.

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





**Figure 1: Land-use zoning of the study area with main sources of BaP and locations of the sampling points. The UTM coordinate system**



**Figure 2: The critical loads of BaP (ng·m⁻²·day⁻¹) plotted against its degradation intensities (%) and exposure time (years) for the different land-use zones in the Eastern Administrative District of Moscow. The land-use zone abbreviations are explained in Table 1.**



**Table 1: BaP contents in the solid fraction of snow and BaP fallout intensities in the winter period for the background territory and the land-use zones of Moscow EAD (2010).**

| Indices | B (10) | Land-use zones[a] (number of samples) | | | | | | A (3) | R (6) |
|---|---|---|---|---|---|---|---|---|---|
| | | T (13) | I (5) | residential areas | | | | | |
| | | | | L (5) | M (13) | H (5) | | | |
| BaP content in the solid fraction of snow | | | | | | | | | |
| Average, ng·g$^{-1}$ | 169 | 3605 | 5732 | 604 | 1225 | 224 | | 197 | 154 |
| Min–max | 69.4–489 | 26.7–22530 | 275–22040 | 232–1466 | 13.2–7278 | 14,5–466 | | 149–238 | 14.2–379 |
| $C_v$, % | 106 | 201 | 163 | 84,8 | 178 | 77.8 | | 22.9 | 90.2 |
| EF | 1.0 | 21.3 | 33.9 | 3.57 | 7.25 | 1.32 | | 1.16 | 0.91 |
| BaP fallout intensity with the solid fraction of snow onto the soil surface | | | | | | | | | |
| Average, ng·m$^{-2}$ per day | 2.0 | 163 | 189 | 18.7 | 43.2 | 7.2 | | 3.4 | 2.4 |
| Min–max | 0.12–0.54 | 0.18–115 | 0.89–49.5 | 0.03–2.14 | 0.03–28.3 | 0.03–2.14 | | 0.24–0.42 | 0.03–0.48 |
| $C_v$, % | 92.1 | 193 | 125 | 102 | 190 | 116 | | 27.0 | 80.2 |
| LF | 1.0 | 80.7 | 93.4 | 9.24 | 21.4 | 3.56 | | 1.66 | 1.18 |
| BaP content in the background and urban soils | | | | | | | | | |
| Average, ng·g$^{-1}$ | 4.93 | 566 | 1563 | 84.3 | 283 | 34.6 | | 4.7 | 164 |
| Min–max | 0–15 | 1.4–3278 | 299–3611 | 1.3–299 | 7.0–1273 | 0–67.0 | | 0–14.0 | 0–798 |
| $C_v$, % | 92.7 | 153 | 83.8 | 147 | 131 | 85.6 | | 171 | 194 |
| EF | 1.0 | 115 | 317 | 17.1 | 57.4 | 7.02 | | 0.95 | 33.3 |

[a] Here and further (Tables 2, 3 and Fig. 2) land-use zones are denoted as: B – background, T – traffic, I – industrial, H,

5   M, L – residential areas with high-, mid- and low-rise buildings respectively, A – agricultural and R – recreational





**Table 2: BaP content in the topsoils in some cities of the world.**

| City | Number of samples | Method of analysis[a] | BaP concentration, ng·g$^{-1}$ | | | Source |
|---|---|---|---|---|---|---|
| | | | Mean | Min | Max | |
| Moscow, EAD (Russia) | 50 | 1 | 409 | 0 | 3611 | Authors' data |
| Moscow (Russia) | 40 | 3 | 115 | 2.1 | 840 | Agapkina et al., 2007 |
| St. Petersburg, Vasilyevsky Island (Russia) | 27 | 3 | 237 | 21 | 1004 | Lodygin et al., 2008 |
| Minsk (Belarus) | 22 | 3 | 45 | 10 | 3100 | Kukharchyuk et al., 2013 |
| Chicago (USA) | 57 | 2 | 2307 | 39 | 17000 | Kay et al., 2008 |
| Glasgow (Scotland) | 20 | 3 | 971 | 132 | 3627 | Morillo et al., 2007 |
| Torino (Italy) | 20 | 3 | 229 | 14 | 3170 | -//- |
| Ljubljana (Slovenia) | 21 | 3 | 77 | 18 | 186 | -//- |
| Bratislava (Slovakia) | 61 | 3 | 195 | 3 | 1200 | Hiller et al., 2015 |
| Miami, Residential Areas (USA) | 21 | 2 | 185 | n/a[b] | n/a | Banger et al., 2010 |
| Tallinn (Estonia) | 39 | 1 | 93 | 4 | 542 | Nikiforova et al., 1993 |
| -//- | 41 | 3 | 156 | n/a | n/a | Trapido, 1999 |
| Shanghai (China) | 55 | 2 | 119 | 23 | 824 | Jiang et al., 2009 |
| Seville (Spain) | 41 | 3 | 67 | 8 | 378 | Morillo et al., 2008 |
| Beijing and its Suburbs (China) | 47 | 2 | 55 | 5 | 270 | Ma et al., 2005 |
| Chang Mai (Thailand) | 30 | 3 | 22 | 7 | 54 | Amagai et al., 1999 |
| Hong Kong (China) | 138 | 2 | 28 | 0 | 1550 | Chung et al., 2007 |
| Tronheim (Norway) | 75 | n/a | 20 | 10 | 2700 | Ottesen et al., 2008 |
| Bergen (Norway) | 435 | n/a | 180 | 5 | 9900 | -//- |

[a] Methods of analysis: 1 – low-temperature spectrofluorimetry, 2 – gas chromatography, mass spectrometry, 3 – liquid chromatography, spectrofluorimetry

5         [b] n/a– not available





**Table 3: The number of years required to lower the BaP contents in soils of EAD to the MPC (20 ng·g$^{-1}$) at the existing fallout levels.**

| Land-use zones[a] | BaP degradation intensity per year[b] | | | | | | | | | |
|---|---|---|---|---|---|---|---|---|---|---|
| | 1 % | 2 % | 3 % | 4 % | 5 % | 6 % | 7 % | 8 % | 9 % | 10 % |
| T | –[c] | – | 152 | 102 | 78 | 63 | 53 | 46 | 40 | 36 |
| I | – | – | 202 | 133 | 101 | 81 | 68 | 59 | 52 | 46 |
| L | – | – | 97 | 56 | 40 | 31 | 25 | 22 | 19 | 17 |
| M | – | – | – | – | – | 90 | 60 | 48 | 40 | 34 |
| H | 59 | 28 | 19 | 14 | 11 | 9 | 8 | 7 | 6 | 6 |
| A | – | – | – | – | – | – | – | – | – | – |
| R | 214 | 106 | 70 | 53 | 42 | 35 | 30 | 26 | 23 | 21 |

[a] Land-use zone abbreviations are explained in Table 1

[b] The BaP degradation intensity is shown as a percentage of its initial concentration in 2010

[c] "–" means that at the accepted BaP degradation intensity, the drop of its concentration below the MPC is impossible