# Peer review of "Benzo(a)pyrene in Urban Environment of Eastern Moscow: Pollution Levels and Critical Loads"

_Atmospheric Chemistry and Physics, 2016_

## Referee Comment (RC1) · Anonymous Referee #2 · 2 Nov 2016

This paper appears a relevant contribution to the PEEX special issue. The paper is relatively well-written and organized. The study appears to be scientifically sound, even though this is a bit difficult to judge in lack of a description of a data quality control (see major comment below). I have a few, mostly minor, comments that should be considered carefully before publication of this paper.

My only major comments on the paper is that something should be said about the quality of the data used in calculating the various numbers presented in the paper, including the associated uncertainties. If such an analysis has been presented in some earlier publication, that should be cited and perhaps the main results summarized very shortly.

The flow of the text at the end of section 2.3 on page 4 could be slightly improved. First,

line 11: . . .is calculated as the difference D - . . .Second, please write . . .where m is the mass of solid matter in a snow sample (ng), n is the number of. . .etc.

Units (lines 2-7 on page 7). I would recommend using the unit g/cm3 instead of t/m3 for density. Please also add the units (mg) into parenthesis on line 6-7.

The flow of the text around equations 1-4 on pages 7-8 should be improved. First, there is no need to write Eq. n before the equations (e.g. the text before eq. 1 could read: . . .can be described as follows:, before eq. 2 as . . .this equation gives, ect.). Second, there is no need to repeat the term alpha1 after eq. 1 because is has already been defined in eq. 1. Third, it is unclear to me why some of the equations are numbered and some others are not. Finally, I do not understand why some equaltion-like things are in the text, while are presented as separate equation in their own line.

Grammatical corrections:

p. 1, line 19, . . ., at the annual. . .

p. 5, line 13: . . .-80), however. . . (add comma)

p. 6, line 2: . . .layers, the. . . (add comma)

p. 6, line 7: . . .typical for cases where relatively. . .

p. 6, line 9: . . .the BaP content

p. 6, line 15: . . .soils, especially. . . (add comma)

p. 8, line 8: . . .in Wild and Jones (1995), . . .

p. 8, line14: . . . exposure times. . .

p. 8, line 15: . . .cannot drop. . .

p. 8, line 31: what is the mening of "there" in this sentence?

p. 9, line 5: . . .cannot exceed. . .

[Figure]

Finally, the authors use both present and past tense when discussing the results. This is mostly correct, yet in some place a present tense is used even though past tense would be more appropriate. Please check out and correct.

---

## Author Comment (AC1) · 3 Dec 2016

Authors are grateful to the reviewer on scrutiny of article and a number of the valuable questions and remarks which allowed to improve the text of article. We added in the section 2.3 Laboratory technique and analysis (p. 4) some details about the analytical method and its precision for BaP, and also about the precision of balance calculations, which were performed using averaged data for land-use zones, so the accuracy of the balance components was higher in 2-3 times as compared to the single samples. This provided sufficient reliability of BaP critical load values.

All editorial guidelines for the text at the end of section 2.3 on page 4 took into account. We changed the unit t/m3 for density to g/cm3 and also added the units (mg) into parenthesis on line 6-7 (page 7). The flow of the text around equations 1-4 on pages

7-8 was improved according the recommendations of the referee. We numbered only the resulted equations and formulae. For brevity, most of the intermediate calculations were given as a text. Some intermediate results are presented in their own line. All necessary grammatical corrections listed by the reviewer are made. We also checked the section with results and changed a present tense in some places to the past tense (p. 4, line 29; p. 5, lines 4, 16, 28, 29, 31, 32; p. 6, lines 10, 20, 21, 22, 23, 25; p. 7, line 11).

———————————————

---

## Referee Comment (RC2) · Dr.Sc. Khaustov (Referee) · 26 Dec 2016

The theme of the paper is within the scope of ACP Journal; it is devoted to the scientific questions, extremely actual but not enough studied in Russia and in other countries. The authors examine the fate and behavior of benzo(a)pyrene (BaP) in urban environment which belongs to very dangerous organic pollutants. The main part of BaP research in urban landscapes is devoted to determining its possible sources and its content in the soils of various land-use areas. Much less publications concern the rate and duration of BaP decomposition in different soil-geochemical conditions that depend on the alkaline-acid and redox conditions, humus content.

In this paper, the first attempt was made to examine all characteristics of BaP balance in urban soils of the Eastern District of Moscow and to calculate the possible

self-purification of urban soils from BaP. The first estimation of BaP critical loads on urban soils was done by authors what emphasize importance and novelty of this article. Previously, this approach was used to calculate the permissible loads of acidity on forest ecosystems in Europe (ICP-Forests) and the critical loads of heavy metals and metalloids (de Vries et al., 1997).

The study area is characterized by various anthropogenic impacts on different land-use zones: industrial areas, highways, residential blocks of different height and density, recreational and post-agrogenic zones. For this area the BaP input with atmospheric deposition and content in the soils were defined, their spatial heterogeneity is shown on maps of atmospheric deposition intensity and technogenic anomalies in the topsoils in the Moscow's Eastern District. Because of poor knowledge of BaP degradation processes multiple calculations are carried out covering the various range of degradation intensity and exposure time data, as well as the initial BaP content in the soils. This paper presents the results of high practical significance: calculations of the dynamics of soils self-purification from polycyclic organic compounds, specifically BaP. They are based on the own large-scale approach of authors applied in Eastern part of Moscow and meet the modern level of studies in the environmental geochemistry. The first application experience of the critical loads approach for BaP showed the significance of investigations of BaP losses through the decomposition including biodegradation.

Touching author's response to the comments of Anonymous Referee #2 to the manuscript, all used methods are valid and clearly outlined. The description of experiments and presented algorithm of calculations is sufficiently complete and precise; in case of interest the experimental part can be reproduced by fellow scientists. The abstract provides a concise and complete summary; it reflects the ground, main algorithms and conclusions of the research. The paper is well structured; the logic of the material is clear. The authors analyzed the vast amount of literature devoted to BaP concentrations in urban environment (e.g. in soils and atmospheric precipitation) including articles in Russian what allow researchers from different countries to get acquainted with this data. The conclusions of the article are of high practical significance. They can be used for planning remediation measures or choosing possible directions in territory use. The results are presented clear enough and allow to give an interpretation and to formulate a conclusion. The original contribution is presented

---

## Author Comment (AC3) · 7 Jan 2017

We are grateful to the reviewer for a detailed interactive comment and high evaluation of the Manuscript. As a result of repeated check of the manuscript text and all numerical data, we found some typos in line "Min–max" of Table 1, section "BaP fallout intensity with the solid fraction of snow onto the soil surface". All values have been corrected (multiplied by 10). Besides, the order of authors has been changed according to alphabetic order of authors' surnames: Nikolay S. Kasimov, Natalia E. Kosheleva, Elena M. Nikiforova, Dmitry V. Vlasov. We also checked the text of Manuscript and made some minor changes listed below.

List of Author's Changes in Manuscript:

1. p. 1, line 3: the order of authors has been changed into "Nikolay S. Kasimov, Natalia

[Figure]

E. Kosheleva, Elena M. Nikiforova, Dmitry V. Vlasov".

2. p. 4, line 19: symbol "-" has been changed into "–".

3. p. 7, line 6: the units "t•m-3" has been changed into "g•cm-3".

4. p. 8, line 18: symbol "-" has been changed into "–".

5. p. 17, table 1, line "Min–max" in Table 1 of section "BaP fallout intensity with the solid fraction of snow onto the soil surface": all values have been multiplied by 10.

Please also note the supplement to this comment:
http://www.atmos-chem-phys-discuss.net/acp-2016-649/acp-2016-649-AC3-supplement.pdf

―――――――――――――

[Figure]

**Supplement:**

[revised manuscript text omitted]

---

## Author Response (AR1)

**Author's response**

**(1) comments from referees/public**

**Anonymous Referee #2** (2 November 2016) has a few comments:

1. My only major comments on the paper is that something should be said about the quality of the data used in calculating the various numbers presented in the paper, including the associated uncertainties. If such an analysis has been presented in some earlier publication, that should be cited and perhaps the main results summarized very shortly.

2. The flow of the text at the end of section 2.3 on page 4 could be slightly improved. First, line 11: … is calculated as the difference D - … Second, please write … where m is the mass of solid matter in a snow sample (ng), n is the number of … etc.

3. Units (lines 2-7 on page 7). I would recommend using the unit g/cm3 instead of t/m3 for density. Please also add the units (mg) into parenthesis on line 6-7.

4. The flow of the text around equations 1-4 on pages 7-8 should be improved. First, there is no need to write Eq. n before the equations (e.g. the text before eq. 1 could read: … can be described as follows:, before eq. 2 as … this equation gives, ect.). Second, there is no need to repeat the term alpha1 after eq. 1 because is has already been defined in eq. 1. Third, it is unclear to me why some of the equations are numbered and some others are not. Finally, I do not understand why some equaltion-like things are in the text, while are presented as separate equation in their own line.

5. Grammatical corrections:
p. 1, line 19: … at the annual …
p. 5, line 13: …-80), however … (add comma)
p. 6, line 2: … layers, the … (add comma)
p. 6, line 7: … typical for cases where relatively …
p. 6, line 9: … the BaP content
p. 6, line 15: … soils, especially … (add comma)
p. 8, line 8: … in Wild and Jones (1995), …
p. 8, line14: … exposure times …
p. 8, line 15: … cannot drop …
p. 8, line 31: what is the mening of "there" in this sentence?
p. 9, line 5: … cannot exceed …

6. Finally, the authors use both present and past tense when discussing the results. This is mostly correct, yet in some place a present tense is used even though past tense would be more appropriate. Please check out and correct.

**Alexander Khaustov** (26 Dec 2016) doesn't have any remarks.

**(2) author's response**

**to Anonymous Referee #2 interactive comment**

1. We added in the section 2.3 Laboratory technique and analysis (p. 4) some details about the analytical method and its precision for BaP, and also about the precision of balance calculations, which were performed using averaged data for land-use zones, so the accuracy of the balance components was higher in 2-3 times as compared to the single samples. This provided sufficient reliability of BaP critical load values.

2-6. All editorial guidelines for the text at the end of section 2.3 on page 4 took into account. We changed the unit t/m3 for density to g/cm3 and also added the units (mg) into parenthesis on line 6-7. The flow of the text around equations 1-4 on pages 7-8 was improved according the recommendations of the referee. We numbered only the resulted equations and formulae. For brevity, most of the intermediate calculations were given as a text. Some intermediate results are presented in their own line. All necessary grammatical corrections listed by the reviewer are made. We also checked the section with results and changed a present tense in some places to the past tense (p. 4, line 29; p. 5, lines 4, 16, 28, 29, 31, 32; p. 6, lines 10, 20, 21, 22, 23, 25; p. 7, line 11).

**to Alexander Khaustov interactive comment**

As a result of repeated check of the manuscript text and all numerical data, we found some typos in line "Min–max" of Table 1, section "BaP fallout intensity with the solid fraction of snow onto the soil surface". All values have been corrected (multiplied by 10). Besides, the order of authors has been changed according to alphabetic order of authors' surnames: Nikolay S. Kasimov, Natalia E. Kosheleva, Elena M. Nikiforova, Dmitry V. Vlasov. We also checked the text of Manuscript and made some minor changes.

**(3) author's changes in manuscript**

For detailed information about changes in manuscript please see a List of Author's Changes in Manuscript below.

**List of Author's Changes in Manuscript**

In all Manuscript: symbol "·" in units has been deleted everywhere.
In all Manuscript: symbol "-" in units has been changed into "–" everywhere.

1. Page 1, line 3: the order of authors has been changed into "Nikolay S. Kasimov, Natalia E. Kosheleva, Elena M. Nikiforova, Dmitry V. Vlasov".
2. Page 1, line 19: "the" has been added before the statement "annual degradation".
3. Page 4, line 7: the statement ", with precision of ±25 %" has been deleted.
4. Page 4, line 7: the statement "Cold extraction of BaP is carried out by n-hexane, its concentration is determined by gas chromatography-mass spectrometry (HPGP-MS), with precision of 15–20 % (Gennadiev and Pikovskii, 1996)." has been added after reference "(Alekseeva and Teplitskaya, 1981)".
5. Page 4, end of chapter 2.3: the statement "The balance calculations were performed using averaged data for land-use zones, so the accuracy of the balance components was higher in 2-3 times as compared to the single samples. This provided sufficient reliability of BaP critical load values." has been added.
6. Page 5, line 3: the form of verb "show" has been changed into "showed".
7. Page 5, line 8: the form of verb "is" has been changed into "was".
8. Page 5, line 17: comma before the phrase "however the difference" has been added.
9. Page 5, line 20: the form of verb "rises up" has been changed into "rose up".
10. Page 5, line 32: the form of verb "show" has been changed into "showed".
11. Page 5, line 33: the form of verb "have" has been changed into "had".
12. Page 5, line 33: the form of verb "is" has been changed into "was".
13. Page 6, line 2: the form of verb "are" has been changed into "were".
14. Page 6, line 3: the form of verb "is" has been changed into "was".
15. Page 6, line 6: comma after the phrase "According to the BaP enrichment of the surface layers" has been added.
16. Page 6, line 11: the statement "This situation is typical for the case, when" has been changed into "This situation is typical for cases where".
17. Page 6, line 13: "the" has been added before the statement "BaP content".
18. Page 6, line 14: the form of verb "are" has been changed into "were".
19. Page 6, line 19: comma before the phrase "especially in the industrial and traffic zones" has been added.
20. Page 6, line 24: the form of verb "exceeds" has been changed into "exceeded".
21. Page 6, line 24: the form of verb "are" has been changed into "were".
22. Page 6, line 25: the form of verb "is" has been changed into "was".
23. Page 6, line 26: the form of verb "is" has been changed into "was".
24. Page 6, line 27: the form of verb "is" has been changed into "was".
25. Page 6, line 29: the form of verb "are" has been changed into "were".
26. Page 7, line 6: the units "t·m-3" has been changed into "g·cm-3".
27. Page 7, line 7: the units "t·m-3" has been changed into "g·cm-3".
28. Page 7, line 8: the units "t·m-3" has been changed into "g·cm-3".
29. Page 7, line 9: the units "mg" has been deleted.
30. Page 7, lines 10-11: the units "mg" has been added.
31. Page 7, line 15: the form of verb "exceed" has been changed into "exceeded".
32. Page 7, line 18: the statement "by an Eq. (1):" has been changed into "as follows".
33. Page 7, line 26: the statement "Eq. (2):" has been deleted.
34. Page 7, line 28: the statement "where $\alpha_1 = e^{-\beta \cdot \alpha_0}$ " has been deleted.
35. Page 8, line 5: the statement "Eq. (3)" has been changed into "formula".
36. Page 8, line 11: the statement "(Wild and Jones, 1995)" has been changed into "Wild and Jones (1995)".
37. Page 8, line 17: the statement "long exposure time" has been changed into "long exposure times".
38. Page 8, line 18: symbol "-" has been changed into "–".

39. Page 8, line 18: the statement "can't" has been changed into "cannot".

40. Page 8, line 28: the statement "Eq. (4)" has been changed into "follows".

41. Page 9, line 4: the statement "there" has been changed into "here".

42. Page 9, line 9: the statement "can't" has been changed into "cannot".

[revised manuscript text omitted]